# Long-haul optical transmission link using low-noise phase-sensitive amplifiers

Samuel L.I. Olsson [1,3], Henrik Eliasson[1], Egon Astra [2], Magnus Karlsson [1] & Peter A. Andrekson [1]

The capacity and reach of long-haul fiber optical communication systems is limited by in-line amplifier noise and fiber nonlinearities. Phase-sensitive amplifiers add 6 dB less noise than conventional phase-insensitive amplifiers, such as erbium-doped fiber amplifiers, and they can provide nonlinearity mitigation after each span. Realizing a long-haul transmission link with in-line phase-sensitive amplifiers providing simultaneous low-noise amplification and nonlinearity mitigation is challenging and to date no such transmission link has been demonstrated. Here, we demonstrate a multi-channel-compatible and modulation-format-independent long-haul transmission link with in-line phase-sensitive amplifiers. Compared to a link amplified by conventional erbium-doped fiber amplifiers, we demonstrate a reach improvement of 5.6 times at optimal launch powers with the phase-sensitively amplified link operating at a total accumulated nonlinear phase shift of 6.2 rad. The phase-sensitively amplified link transmits two data-carrying waves, thus occupying twice the bandwidth and propagating twice the total power compared to the phase-insensitively amplified link.

[1] Department of Microtechnology and Nanoscience, Chalmers University of Technology, SE-412 96 Gothenburg, Sweden. [2] Thomas Johann Seebeck Department of Electronics, Tallinn University of Technology, Tallinn 19086, Estonia. [3]Present address: Nokia Bell Labs, 791 Holmdel Road, Holmdel, NJ 07733, USA. Correspondence and requests for materials should be addressed to S.L.I.O. (email: samuel.olsson@nokia-bell-labs.com)

The achievable transmission performance of fiber optical transmission systems is limited by amplifier noise and fiber nonlinearities degrading the signal[1–3]. Phase-sensitive amplifiers (PSAs) can provide low-noise amplification, because at high gains their noise figure (NF) is 3 dB lower than that of even ideal phase-insensitive amplifiers (PIAs)[4,5]. Using an alternative NF definition where only the signal power is accounted for (idler power is neglected), the NF difference between PSAs and PIAs increases to 6 dB[5]. PSAs are also capable of all-optical mitigation of nonlinear transmission distortions[6–8]. Using PSAs low-noise amplification and nonlinearity mitigation capabilities, PSAs can potentially improve the transmission performance of fiber optical transmission systems[9,10].

PSAs can be realized, for example using, parametric gain in $\chi^{(2)}$ nonlinear materials through three-wave mixing (TWM)[11], or $\chi^{(3)}$ nonlinear materials through four-wave mixing (FWM)[12]. Typically, two weak waves, called signal and idler, are amplified by one or two high-power waves, called pumps. Depending on how the frequencies of the interacting waves are chosen, different amplification schemes are possible. Two common schemes are the one-mode PSAs in which signal and idler are frequency degenerate and the two-mode PSAs in which signal and idler are frequency non-degenerate.

In one-mode PSAs, one quadrature is amplified while the other quadrature is deamplified, squeezing the signal phase along the direction of the amplified quadrature[4]. If the PSA is operated in unsaturated regime, phase noise in the squeezed quadrature will be converted into amplitude noise in the amplified quadrature. If, however, the PSA is operated in saturation both phase and amplitude noise can be suppressed thus making this scheme suitable for simultaneous phase and amplitude regeneration of binary phase-shift keying (BPSK) signals[13–15]. Using this scheme, a two times reach extension, originating from phase and amplitude regeneration, not low-noise amplification, has been demonstrated[16,17]. Two severe drawbacks of the one-mode PSA scheme is that it is inherently single-channel and that it is only suitable for BPSK signals. Using other PSA-based schemes, regeneration of more advanced modulation formats such as quadrature phase-shift keying (QPSK)[18,19], and star 8-quadrature amplitude modulation (QAM)[20], have been demonstrated as well as simultaneous regeneration of more than one channel[21,22].

Another way to benefit from PSAs is to utilize their capabilities of low-noise amplification and nonlinearity mitigation. This can be done using two-mode PSAs implemented with the so-called copier-PSA scheme[23]. Using the copier-PSA scheme, all signal phase states will experience low-noise amplification thus providing modulation-format transparency[24]. Moreover, two-mode PSAs are multi-channel compatible and can be used for amplification of wavelength division multiplexing (WDM) signals[25]. In ref.[26], it was shown that two-mode PSAs potentially can be combined with multi-channel amplitude regenerators for multi-channel regeneration of advanced modulation formats. For details on the requirements regarding the tracking and alignment of polarization in PSA links see ref.[27].

Mitigation of fiber nonlinearities to extend transmission reach is a vivid research area currently[28], and many different schemes have been proposed, e.g., phase conjugated twin waves[29] or conjugate data repetition[30], which are based on the idea that the signal and the conjugate signal are co-propagated through the same medium and coherently superposed to suppress the nonlinear-induced phase distortion. Cancellation of nonlinear distortion by digital signal processing[31] in the receiver[32] or transmitter[33] has also been demonstrated, as has optical phase conjugation (OPC)[34]. Typically, a doubling or at most a tripling of the system reach have been reported by these schemes, at the expense of spectral efficiency and/or complexity. A way to further enhance performance could be to distribute the compensation, which is attractive for all-optical schemes such as PSAs or OPC, and for OPCs that was recently demonstrated[35,36], although relatively moderate Q-factor improvements over single OPCs were reported.

Here we present experimental evidence that in-line PSAs, can provide an unprecedented nonlinear tolerance and transmission reach extension[9,10]. In this demonstration of a recirculating loop (i.e., long-haul) transmission experiment with in-line PSAs, we benefit from the inherent simultaneous low-noise amplification and nonlinearity mitigation. This scheme, which is both modulation format-independent and multi-channel compatible[5], is shown experimentally to have a 5.6 times reach improvement compared to a transmission link using conventional in-line erbium-doped fiber amplifiers (EDFAs) when transmitting a 10 GBd QPSK signal. The accumulated nonlinear phase shift in the PSA link is 6.2 rad, which we believe is the highest nonlinear tolerance ever reported in a lumped-amplifier system. These results demonstrate not only the feasibility of realizing long-haul transmission links using low-noise PSAs but also significant improvement over conventional approaches. The concept of amplification using cascaded PSAs might also find applications in the field of quantum information science, where generation and processing of quantum states are of interest.

## Results

**Basic principle.** The amplifier implementation we consider in this work is the degenerate pump, two-mode PSA. It consists of three waves, an intense pump surrounded by a signal and an idler. The input–output relation for the signal and idler is given by

$$\begin{pmatrix} u_s \\ u_i^* \end{pmatrix}_{out} = \begin{pmatrix} \mu & \nu \\ \nu^* & \mu^* \end{pmatrix} \begin{pmatrix} u_s + n_s \\ u_i^* + n_i^* \end{pmatrix}_{in} \qquad (1)$$

where $u_{s,i}$ are the signal and idler wave amplitudes, $n_{s,i}$ represents vacuum noise present at the input, and the amplifier is characterized via the scalar coefficients $\mu$ and $\nu$, where $|\mu|^2 - |\nu|^2 = 1$ ensures photon-number conservation, i.e., two pump photons are converted into one signal and one idler photon. If the input idler wave is absent, $u_{i,in} = 0$ then the output signal is amplified phase insensitively with gain $G_{PIA} = |\mu|^2 \approx |\nu|^2$, where the approximate equality holds in the limit of high gain.

In our experiment, we employ a sequence of these amplifiers with intermediate fiber losses that are compensated for by the provided gain. The first amplifier has $u_{i,in} = 0$, so it copies the conjugate incoming signal to the output idler wave. The generated signal-idler pair then propagates through all subsequent amplifiers, while achieving a phase-sensitive gain $G_{PSA}$ of approximately $4G_{PIA}$ due to coherent addition of signal and idler conjugate.

In contrast to the signal, for which the gain is 6 dB higher in phase-sensitive (PS) mode than in phase-insensitive (PI) mode, the gain for the vacuum noise is always $2G_{PIA}$ since the noise is uncorrelated between signal and idler and will thus not add coherently. By comparing PI- and PS-operation at the same signal gain the difference between PIA and PSA amplification can be stated as that a PSA will add 6 dB less noise than a PIA. The first 3 dB of this improvement comes from the phase-sensitive nature of the gain, which releases the PSA from being constrained by the 3 dB quantum limit on PIA NF[4], at the expense of using half of the available bandwidth for propagating the idler[37]. This NF improvement has been characterized in detail in refs.[38,39] The second 3 dB of the improvement comes from the fact that the data in the two-mode-PSA-amplified link are carried by two beams (signal and idler) of equal powers, which makes the effective total data-carrying power in the PSA link twice that of

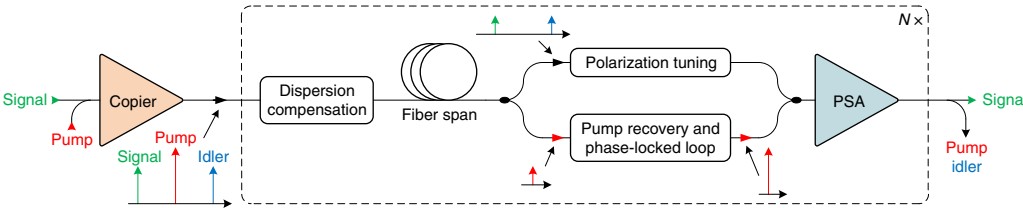

**Fig. 1** Long-haul PSA-amplified link. Conceptual schematic of a long-haul PSA-amplified link implemented using the copier-PSA scheme

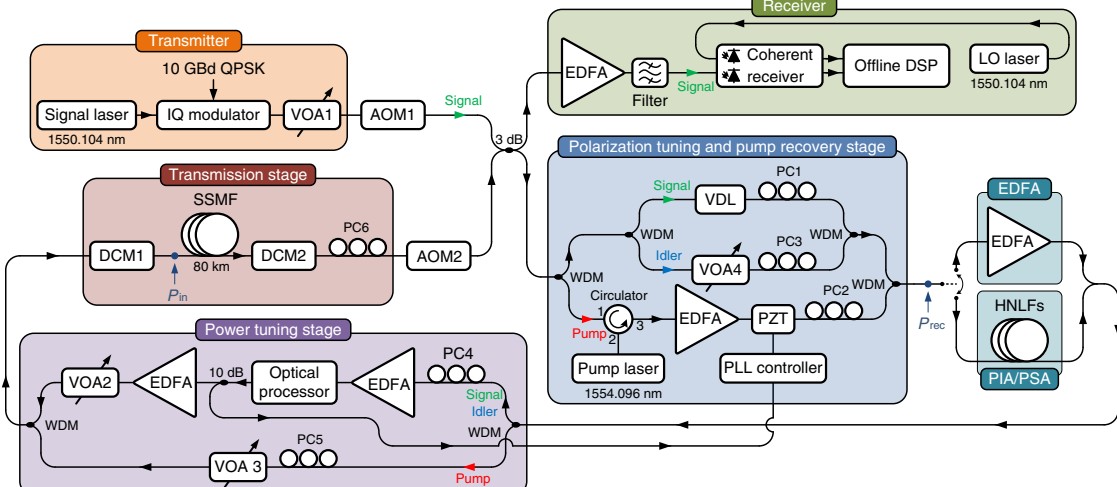

**Fig. 2** Experimental set-up. Recirculating loop set-up used to demonstrate long-haul transmission with in-line PSA-based amplification. Option with in-line EDFA- and PIA-based amplification used for benchmarking is also shown. Colored arrows indicating waves represent PSA case for the second and the following round trips. IQ modulator in phase/quadrature modulator, AOM acousto-optic modulator, VDL variable delay line, PC polarization controller, HNLF highly nonlinear fiber, PLL phase-locked loop, DSP digital signal processing, LO local oscillator

the PIA link[37]. Described above is the so-called copier-PSA scheme, and its linear link properties were analyzed in refs.[40,41] and experimentally verified for a single-span link in refs.[5,42] A conceptual schematic of a multi-span implementation of the copier-PSA scheme is shown in Fig. 1.

**Experimental set-up**. The experimental set-up used to demonstrate long-haul transmission with in-line PSAs is illustrated in Fig. 2. A signal modulated with 10 GBd QPSK data were launched into a recirculating loop. During the first round trip ($N = 1$), only one wave, the signal, was present at the input of the polarization tuning and pump recovery stage and a pump wave was generated using a laser. After combining the signal with the pump using a WDM coupler, the two waves were launched into a fiber optical parametric amplifier (FOPA) where a conjugated copy of the signal, the idler, was generated. During the first round trip, the FOPA thus operated as a copier. The three waves were then passed through a power tuning stage where an optical processor (OP) was used to filter the signal and idler as well as adjust their powers. Following the OP, the signal and idler were passed through an EDFA followed by a variable optical attenuator (VOA) for launch power tuning. The pump was passed through a separate path and was attenuated using a VOA. The transmission link consisted of two tunable fiber Bragg-grating dispersion compensating modules (DCMs) and an 80 km standard single-mode fiber (SSMF) transmission span. The combined loss of the SSMF span and the second DCM was 21.5 dB.

During the second and the following round trips ($N \geq 2$), the pump was regenerated in the polarization tuning and pump recovery stage by injection-locking it to the pump laser and

subsequently amplifying it with an EDFA. The process of self-injection-locking enabled stable injection-locking over many circulations. The signal and idler were split into two separate paths and the delay between them introduced by the SSMF was compensated for. A phase-locked loop based on a piezoelectric transducer (PZT) fiber stretcher was used to compensate for any dynamic phase drifts between the arms introduced by temperature and acoustic influence. After the polarization tuning and pump recovery stage, the waves were launched into the FOPA, which now operated as a PSA with 22 dB net gain providing low-noise amplification and nonlinearity mitigation. To compensate for the fact that the copier operated as a PIA, adding 6 dB more noise to the signal than the following PSAs, the signal power launched into the loop was 6 dB higher than the power present at the point of the loop input after the first round trip. The received power was measured at point $P_{rec}$ and the loss from point $P_{in}$ to point $P_{rec}$ was 39 dB. In each round trip, part of the light was coupled out of the recirculating loop and detected using a coherent receiver. A more detailed description of the experimental set-up is presented in the Methods section.

**Constellation diagrams**. To benchmark the performance of the PSA-amplified link, measurements were also performed on an EDFA-amplified link and a FOPA-PIA-amplified link. The FOPA-PIA-amplified link was obtained by blocking the idler in the OP and fully attenuating the pump in the power tuning stage. The EDFA link was obtained by replacing the FOPA with an EDFA and turning off the pump laser as well as the pump booster EDFA. The three cases were compared both by studying constellation diagrams and by measuring bit error rate (BER).

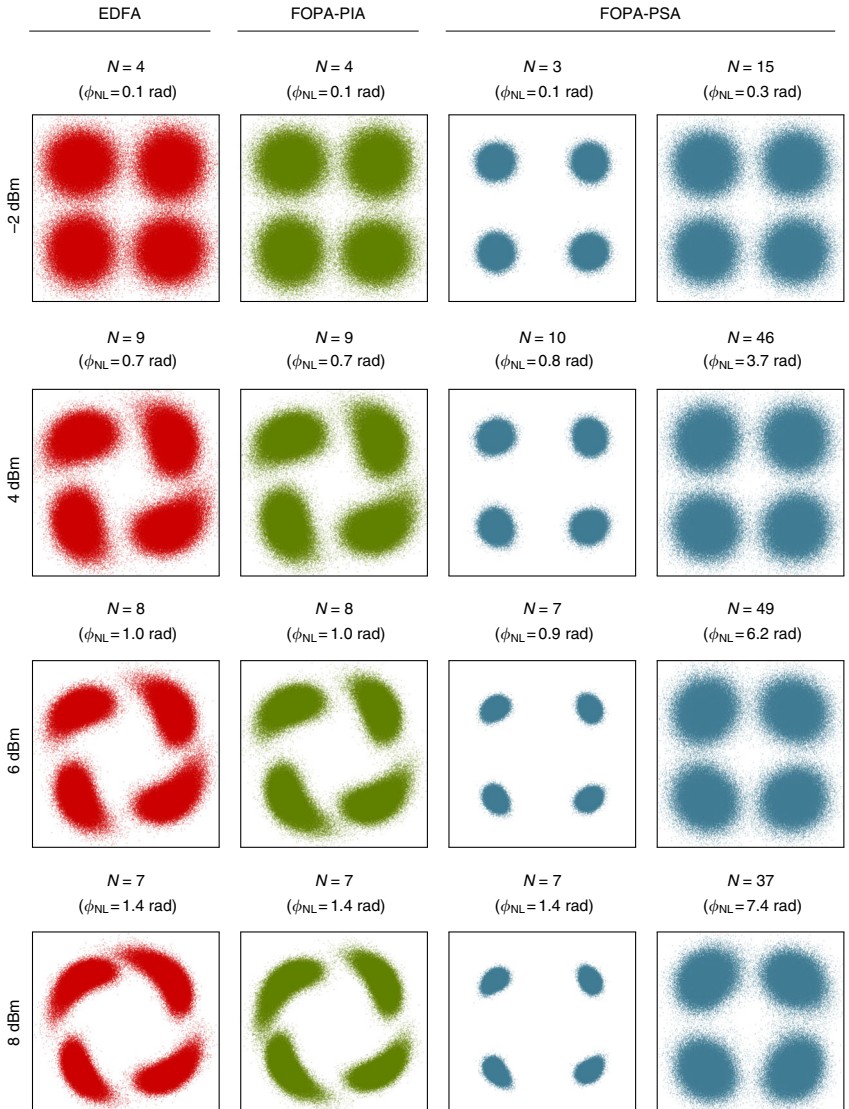

**Fig. 3** Signal constellation diagrams. Constellation diagrams at various launch powers using EDFA-based amplification, FOPA-PIA-based amplification, and FOPA-PSA-based amplification. Constellations in column one, two, and four corresponds to a BER of $10^{-3}$. The variable $N$ indicates the number of round trips where each round trip include an 80 km dispersion compensated SSMF span and $\phi_{NL}$ denotes the accumulated nonlinear phase shift

Figure 3 shows constellation diagrams at various launch powers (measured at point $P_{in}$) for the three investigated amplification schemes. The constellations in columns one, two, and four correspond to a measured BER of $10^{-3}$ while the third column shows the constellations for the FOPA-PSA case at the closest available number of round trips to the EDFA case. The variable $N$ indicates the number of round trips. The accumulated nonlinear phase shift was calculated using $\phi_{NL} = \gamma P_{in} L_{eff}$, where $\gamma$ is the nonlinear coefficient, $P_{in}$ is the launch power, and $L_{eff}$ is the effective length defined as $L_{eff} = [1 - \exp(-\alpha L)]/\alpha$ with $\alpha$ being the fiber attenuation and $L$ the link length, and is shown in parenthesis above each constellation. When calculating $\phi_{NL}$ we used $\gamma = 1.5\,\mathrm{W}^{-1}\,\mathrm{km}^{-1}$, $\alpha = 0.2\,\mathrm{dB\,km}^{-1}$, and $L = 80$ km.

It is clear from Fig. 3 that EDFA- and FOPA-PIA-based amplification provide similar performance from $-2$ dBm launch power, where the reach is limited by amplifier noise, up to 8 dBm launch power, where reach is limited by nonlinear distortions. We can also see that PSA-based amplification significantly reduces the accumulated amplifier noise as well as the impact of fiber nonlinearities, thus allowing for improved reach at all launch powers.

**Bit error rate measurements**. Measured BER versus number of round trips and transmission distance at various launch powers is presented in Fig. 4a. It can be seen that the EDFA case and FOPA-PIA case are close to indistinguishable while the PSA case shows significantly improved reach. The reach improvement as well as the maximum number of round trips (for a BER of $10^{-3}$) versus launch power is presented in Fig. 4b. From the figure we note that the optimal launch power for the PSA case is 6 dBm while the optimal launch power in the EDFA- and FOPA-PIA cases is 4 dBm. At 6 dBm launch power, the reach improvement using PSA-based amplification is about six times while if the comparison is made at optimal launch powers, the reach improvement is 5.6 times.

## Discussion
Our demonstration of long-haul PSA-amplified transmission was performed using a signal with 10 GBd QPSK data. However, in principle any modulation format and symbol rate can be used with the copier-PSA scheme. Increasing the symbol rate will make it more challenging to achieve good enough temporal

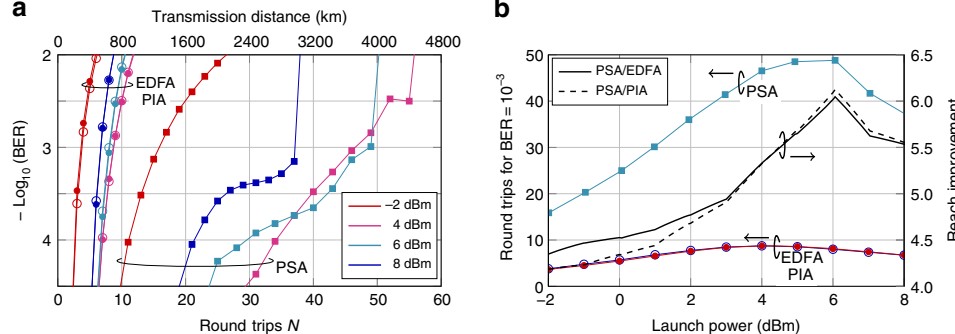

**Fig. 4** BER characterization. **a** BER curves at various launch powers using in-line EDFA- (filled circles), FOPA-PIA- (open circles), and FOPA-PSA-based (filled squares) amplification. **b** Number of round trips giving a BER of $10^{-3}$ versus launch power and reach improvement comparing EDFA- (filled circles) and FOPA-PIA-based (open circles) amplification to PSA-based (filled squares) amplification

alignment of the waves but by using high precision delay lines this should be possible. The copier-PSA performance in linear regime is not expected to depend on either symbol rate or modulation format. However, the ability to mitigate nonlinearities might depend on both symbol rate and modulation format and will require further investigation. Approaches that have been suggested as means to improve nonlinearity mitigation at, e.g., higher symbol rates in PSA-amplified links are addition of distributed Raman amplification[43] or multi-span dispersion map optimization[44]. In our demonstration, we transmitted a single channel but with the copier-PSA scheme multi-channel transmission is possible although with increased complexity due to required polarization, delay, and phase alignment of each channel.

Using a recirculating loop to demonstrate long-haul PSA-amplified transmission is a good approach to demonstrate the possible performance improvements that can be gained using in-line PSAs. However, using a recirculating loop simplifies certain aspects of the implementation and in order to realize a real transmission link with in-line PSAs a few challenges remain to be solved. One such challenge is the pump recovery. The injection-locking-based pump recovery is sensitive to frequency differences between the incoming pump wave and the free-running pump laser frequency. In our recirculating loop set-up, this frequency difference could be kept small since it was dictated by the pump laser wavelength drift during the measurement time which was 24 ms for 60 round trips. In a real transmission link, a feedback system would be required to tune the frequency of the slave lasers to match the frequency of the incoming pump wave. Another aspect that will be more challenging in a real transmission link is the polarization alignment of the involved waves. In our recirculating loop set-up, this alignment could be done manually. However, in a real transmission-link polarization tracking would be required to align the waves and to keep them aligned over time.

We have demonstrated the possibilities and potential of using cascaded PSAs in the context of high-speed optical communications. However, our results might also find applications in quantum informatics and related fields where generation and processing of quantum states are of interest.

## Methods

**Recirculating loop experiment.** A continuous wave (CW) laser (Keysight N7711A) at 1550.104 nm with <100 kHz linewidth, $-145$ dB Hz$^{-1}$ relative intensity noise (RIN), and 30 mW output power was modulated with 10 GBd QPSK data (pseudorandom bit sequence (PRBS) of length $2^{15}-1$) using a LiNbO$_3$-based single polarization I/Q modulator. The electrical signals driving the I/Q modulator were generated using a bit pattern generator (SHF 12103A) followed by electrical amplifiers (SHF 804 TL). After passing a VOA, VOA1, for loop launch power tuning the signal was passed into a recirculating loop that was controlled using a

loop controller (Brimrose AMM-55-8-70-C-RLS(nfs)-RM) containing two acousto-optic modulators (AOMs).

During the first round trip only the signal was present at the input of the polarization tuning and pump recovery stage and a CW pump wave at 1554.096 nm was generated using a distributed feedback (DFB) laser without isolator (EM4 AA1406-192900-100) with <1 MHz linewidth, $-150$ dB Hz$^{-1}$ RIN, and 100 mW output power. The pump wave was subsequently amplified using a 3 W fanless high-power EDFA (IPG EUA-3K-C-CHM) and attenuated to obtain 1 W at the FOPA input. The signal was combined with the pump before the FOPA using a WDM coupler and the signal and pump state of polarizations (SOPs) were aligned using PC1 and PC2 for maximum FOPA gain. With only signal and pump present at the FOPA input, the FOPA operated as a PIA with 16 dB net gain. The FOPA consisted of four cascaded spools of strained highly nonlinear fiber (HNLF) (OFS HNLF-SPINE with zerodispersion wavelength (ZDW) at 1543 nm) of lengths 101, 124, 156, and 205 m, with in-line isolators placed between the individual spools for stimulated Brillouin scattering (SBS) suppression[45]. During the first round trip, the FOPA generated a conjugated copy of the signal, frequency- and phase-locked to the signal and pump, at the idler wavelength through FWM.

After the in-line amplifier, the waves were led to a power tuning stage where the high-power pump was separated from the signal and idler waves. The signal and idler were amplified using an EDFA (Nortel) and then passed into an OP (Finisar WaveShaper 1000S) for filtering (0.4 nm bandpass filters) and power tuning such that they were balanced in power at point $P_{in}$, just before the transmission fiber. The two waves were then led into a custom built EDFA with 3.1 dB NF and 25 dBm output power followed by VOA2 for launch power tuning. PC4 was tuned so that the polarization dependent loss (PDL) experienced by the signal over the transmission stage was minimized. The pump was attenuated using VOA3 to obtain $-5$ dBm at point $P_{in}$ and PC5 was tuned such that the SOP of the pump launched into the pump laser in the second round trip was aligned with the free-running pump laser SOP.

The transmission link constituted of two 100 GHz channel grid tunable fiber Bragg-grating DCMs (TeraXion TDCMX-C100-($-80$ km/$+5$ km)), DCM1 for dispersion pre-compensation and DCM2 for post-compensation, and an 80 km SSMF transmission span. The dispersion map was experimentally optimized for longest reach in a strongly nonlinear regime (6 dBm launch power). In the PSA case, the optimum dispersion map was 289 ps nm$^{-1}$ pre-compensation and 986 ps nm$^{-1}$ post-compensation. In both the EDFA- and the FOPA-PIA case, the optimum dispersion map was 68 ps nm$^{-1}$ pre-compensation and 1207 ps nm$^{-1}$ post-compensation. The amount of per span residual dispersion was experimentally optimized for longest reach in a nonlinear transmission regime for the PSA case and was <35 ps nm$^{-1}$. This amount of residual dispersion had a negligible impact on the performance in linear transmission regime both for the PSA and PIA cases as well as for the PIA cases in nonlinear transmission regime due to the low symbol rate and few round trips. The launch power was measured as signal power at point $P_{in}$. PC6 was adjusted so that the signal SOP at the beginning of the second round trip was the same as the SOP of the signal launched into the transmission loop. The round trip time was 0.4 ms.

During the second and the following round trips both signal, idler, and pump were present at the input of the polarization tuning and pump recovery stage. The pump was separated from the signal and idler and injection-locked to the pump laser. This process of self-injection-locking enabled stable locking over many circulations. The signal and idler were also separated, and the delay between them introduced by the transmission fiber was compensated for using a variable delay line (VDL) with a 1 dB insertion loss. The idler was attenuated such that the signal and idler had equal power going into the FOPA and their SOPs were aligned using PC1 and PC3 so that the FOPA gain was maximized. A phase-locked loop (PLL) based on a PZT fiber stretcher was used to compensate for any dynamic phase drifts between the arms introduced by temperature and acoustic influence. The FOPA-PSA net gain was 22 dB. For simplicity the PSA-amplified link was implemented such that the same FOPA was used both for the copier and the PSA.

As a consequence of this, the first and last in-line amplifiers in the PSA-amplified link were PIAs. In order to compensate for the extra signal degradation caused by the first in-line PIA, the signal power launched into the loop was 6 dB higher than the power present at the point of the loop input after the first round trip. The absence of nonlinearity mitigation in the last in-line amplifier in the link was not compensated for. Due to the placement of the loop output coupler, the loss of the final span was lower than the other spans by ~2 dB. For the EDFA case and the FOPA-PIA case, this resulted in slightly better performance compared to what would have been achieved in a link in which all spans had the same loss. For the PSA case, the performance was still worse then it would have been if the last amplifier in the link was a PSA. Note, however that the impact of having slightly lower loss in the last span is negligible after many circulations.

In each round trip, part of the light was coupled out of the recirculating loop and amplified by an EDFA (JDS Uniphase OAB optical amplifier) followed by an optical filter (OTF-30M-12S2) with a 3 dB bandwidth of 0.9 nm centered at the signal wavelength. The amplified and filtered signal was then coupled into a coherent receiver (NeoPhotonics Integrated PBS ICR) along with a local oscillator wave generated by a CW laser (IDPhotonics CBDX1-1-C-H01-FA) at the signal wavelength with <100 kHz linewidth, $-145$ dB Hz$^{-1}$ RIN, and 40 mW output power. The signal was sampled at 50 GS s$^{-1}$ using a real-time sampling oscilloscope (Tektronix DPO73304SX) with 33 GHz analog bandwidth. For each round trip, $2.5 \times 10^6$ samples (corresponding to 50 μs at 50 GS s$^{-1}$) were taken in the middle of the 0.4 ms long burst and then post processed off-line using conventional DD-LMS-based digital signal processing (DSP). The back-to-back signal-to-noise ratio (SNR) penalty of the transmitter and receiver was 0.5 dB at a BER of $10^{-3}$.

For the EDFA case, the pump laser was turned off and the FOPA was substituted with a custom built EDFA with 3.1 dB NF and 25 dBm output power followed by a VOA tuned such that the net gain of the EDFA and VOA was 22 dB. For the FOPA-PIA case, the idler was blocked in the OP and the pump was fully attenuated before the transmission stage using VOA3. The in-line FOPA-PIA net gain was 16 dB.

**Data availability**. The data that support the findings of this study are available from the corresponding author upon reasonable request.

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

## Acknowledgements

This work was supported by the European Research Council under grant agreement ERC-2011-AdG-291618 PSOPA, by the Swedish Research Council (VR, Grant No. 2015-00535), and by the Wallenberg Foundation. We would like to thank Abel Lorences-Riesgo for helpful discussions.

## Author contributions

S.L.I.O., H.E., and E.A. designed and built the set-up. S.L.I.O. and H.E. carried out the experiment and data analysis. P.A.A. and M.K. provided overall technical leadership and supervision. S.L.I.O., M.K., and P.A.A. jointly wrote the paper.

## Additional information

**Competing interests:** The authors declare no competing interests.

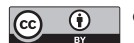

