## [Peer Review File · Nature Communications]

Reviewers' comments:

Reviewer #1 (Remarks to the Author):

The authors report a very thorough and systematic study of performance of an ultra-long-haul 10 Gbaud QPSK link employing phase-sensitive amplifiers (PSAs) in comparison to the links based on conventional phase-insensitive fiber-optic parametric amplifiers (FOPA-PIAs) or EDFAs with nearly quantum-limited noise figures. They demonstrate a very impressive result: the PSA-enabled link is capable of 5.6 times greater reach than the reach of the EDFA- or FOPA-PIA-based link. Even more remarkably, they show that the PSA-based link can tolerate a total accumulated nonlinear phase shift of 6.17 radians, incurred by each of signal and idler beams, which is 6–8 times greater than the nonlinear tolerance of the conventional PIA-based links.

It was previously predicted [e.g., Opt. Express **13**, 7563 (2005)] that the two-mode PSAs can improve the long-haul transmission links by combination of the reduction in the amplified spontaneous emission (ASE) noise and mitigation of nonlinear impairments. The authors present the first experiment that proves this prediction by showing that the net reach increase is noticeably greater than 4 times (the number expected from just the 6-dB ASE noise suppression in a linear propagation regime), i.e., both ASE reduction and nonlinear mitigation are fully realized here. These impressive results can influence the optical communication community to accept the PSAs as an attractive alternative to conventional amplifiers.

The key enablers of these original and impressive demonstrated results are the PSAs – a novel class of amplifiers rapidly gaining interest in the optical communication field. The authors reported preliminary observations of PSA- and PIA-based QPSK long-haul links at a conference [9], and this manuscript presents a much refined, very thorough, and definitive version of the PSA/PIA comparison, which sets a very high bar for proper “apple-to-apple” experimental comparisons between the links employing different amplifier types. These comparisons are extremely challenging, because the two-mode PSA is not just a simple drop-in replacement for an EDFA, especially in a recirculating loop, and I expect that the authors’ outstanding work will serve as a “gold standard” for how such comparative experiments should be carried out in the future.

The authors’ conclusions are very convincing and supported by large amounts of well-presented and clearly-explained data.

I strongly recommend this manuscript to be accepted for publication in Nature Communications. To improve the clarity of the presentation of this very sophisticated experiment, I advise the authors to consider the following suggestions / comments:

1. The authors should be careful in using statements about PSAs capable of adding 6 dB less noise than PIAs (3 occurrences: abstract, 1st paragraph of the introduction, last paragraph of “Basic principles”) without qualifying / clarifying them. For the same signal power gain as a PIA, PSA indeed produces 6 dB less ASE power. This 6-dB advantage, however, is not “real,” i.e., it does not result in

6-dB improvement in the signal-to-noise ratio (SNR) from PIA to PSA for the same amplifier input power. In the case of a signal/idler degenerate (one-mode) PSA, this is because the ASE after the PSA no longer has completely random phase (its phase is aligned with that of the amplified signal, i.e., the PSA quadrature noise is only 3-dB lower than that of the PIA), which leads to detected SNR improvement over PIA to be only 3 dB. In the case of a two-mode PSA, the ASE of the signal still has random phase, but the effective input signal power into the amplifier is doubled compared to the PIA, because of the presence of data-carrying idler beam – that is, for the same total input power as the PIA the two-mode PSA yields only 3-dB better SNR. Hence, in the two-mode PSA the second 3 dB of the 6-dB noise reduction is an artefact of propagating twice as much total power in the transmission link. Thus, I believe the reader is misled when the manuscript mentions 6-dB noise reduction without adding that it requires propagating twice as much total power in the transmission fiber. While PSA experts can understand what the authors mean, the general Nature Communications audience will be very confused by 6-dB noise reduction (as opposed to 3-dB reduction well-known from quantum mechanics [4]) without a proper mentioning of the corresponding power increase. Here are the suggested modifications for the three occurrences of “6 dB less noise”:

- a. Abstract: a good way to explain what is responsible for the 6-dB less noise is by modifying the third sentence from the end of the abstract into “The phase-sensitively amplified link transmits two data-carrying waves, thus occupying twice the bandwidth **and propagating twice the total power** compared to the phase-insensitively amplified link.” (boldface is the added modification)
- b. In the second sentence of the introduction, “6 dB less noise” is very ambiguous, because neither “noise” nor “gain” is defined here. As discussed above, the 6-dB figure artefact comes from omitting the proper definition of either gain (not accounting for increased input power) or noise (not considering that all amplified noise power is concentrated in one quadrature). Citing Ref. [4] in this context is also misleading, because this reference never discussed more than 3 dB of PSA advantage over PIA. Thus, to avoid any confusion and controversy, the authors should re-write the second sentence of the introduction in terms of the PSA noise figure, on which everyone agrees that it is lower than the ideal PIA noise figure by 3 dB: “Phase-sensitive amplifiers (PSAs) can provide low-noise amplification, **because at high gains their noise figure is 3 dB lower and compared to than that of even ideal phase-insensitive amplifiers (PIAs) operating at the same gain, PSAs add 6 dB less noise to the signal** [4], [5].”
- c. In the last paragraph of “Basic principles,” after the sentence ending with “...PSA will add 6 dB less noise than a PIA.” an explanation needs to be given, along the lines of “**First 3 dB of this improvement comes from the phase-sensitive nature of the gain, which releases the PSA from being constrained by the 3-dB quantum limit on PIA noise figure [4], at the expense of sacrificing half of the available bandwidth for**

propagating the idler [*]. This noise-figure improvement has been characterized in detail in [**,***]. The second 3 dB of the improvement comes from the fact that the data in the two-mode-PSA-amplified link is carried by two beams (signal and idler) of equal powers, which makes the effective total data-carrying power in the PSA link twice that of the PIA link [*].” Here Ref. [*] is Opt. Express **13**, 7563 (2005), in which the two-mode PSA and copier-PSA scheme were first proposed and explained, and [**,***] are Opt. Express **18**, 14820 (2010) and Opt. Lett. **36**, 722 (2011).

I would like to stress that the need to clarify the 6-dB improvement does not take anything away from the breakthrough experimental results demonstrated in the paper. Such clarification, in my opinion, is needed to give these results the physical interpretation they deserve, without contradicting quantum mechanics. From a practical perspective, the reader also needs to be made aware that there is “no free lunch”: two-mode PSA requires more bandwidth and more propagating power to achieve 6-dB better noise performance than PIA.

2. I suggest using singular word “link” in the title of the paper.
3. The authors use the term “span-wise nonlinearity mitigation” without explaining what it means. I think something like “**nonlinearity mitigation after each span**” would make it more clear.
4. I would also caution authors against making explicit suggestions (in the abstract and conclusions) of potential use of the PSAs for entangled photon amplification. While PSAs are important for generation of squeezed and continuous-variable entangled states and could find some other potential applications in quantum information sciences, such as that in Ref. [39], they have been proven useless for quantum communications and detrimental to entangled states [“Gaussian-only regenerative stations cannot act as quantum repeaters,” R. Namiki, O. Gittsovich, S. Guha, and N. Lütkenhaus, Phys. Rev. A **90**, 062316 (2014)]. I suggest limiting the authors’ statements to a more general form such as “might also find applications in the field of quantum information science, where generation and processing of quantum states are of interest.”
5. In addition to some of the references above, the manuscript could also benefit from a couple of more references, such as:
 - a. Citing [Science **348**, 1445 (2015)] in the context of “Cancellation of nonlinear distortion by digital signal processing” in the last paragraph of page 1;
 - b. Mentioning in the last sentence of the second-to-last paragraph of page 1 that the multi-channel compatible two-mode PSAs can be potentially combined with multi-channel amplitude regenerators [Nature Comm. **8**, 884 (2017)] for multi-channel regeneration of advanced modulation formats.
6. 4th sentence of the “Discussion” section stating that “The performance in linear regime is not expected to depend on neither symbol rate or modulation format” should be rephrased as “The **copier-PSA** performance in linear regime is not expected to depend on neither symbol rate or modulation format”. Otherwise, “performance” implies BER, which of course will depend on both symbol rate

and modulation format, and keeping same BER would require higher SNRs for higher symbol rates and more advanced formats.

7. A couple of suggestions for the Figures:
 - a. “Colored arrows indicating waves” in Fig. 2 appear too small to be recognized as arrows (can be easily confused with circles denoting WDM couplers and input and received power points) – perhaps they can be enlarged.
 - b. EDFA and PIA data in Fig. 4(a) would benefit from employing different symbols (e.g., filled vs empty circles) to show that there are actually two nearly overlapping data sets for each launch power.
8. There are a number of small details about the experiment that need to be added or clarified, such as:
 - a. What is PRBS15? Is it 2^{15} , $2^{15}-1$ or something else?
 - b. What does it mean “The received power was measured at point P_{rec} ” on line 182 of page 3? Does it mean that the signal going into the receiver EDFA (green box in Fig. 2) was attenuated by amount equivalent to the loss of “Polarization tuning and pump recovery-stage” unit, but this attenuator is not shown in Fig. 2?
 - c. The important fact that in the PSA-based link the signal power launched into the first loop round-trip was 6 dB higher than that going into the second round-trip is not mentioned until the Methods section at the end of the article. I think this information should be provided much earlier – either in “Experimental set-up” or in “Constellation diagrams” section. Otherwise, this lack of information leaves the reader puzzled throughout most of the manuscript about whether the signal power in the SMF in the first loop is 6-dB lower than in the subsequent loops or not.
 - d. Why does the third column of Fig. 3 show “the constellations for the FOPA-PSA case at the closest available number of round trips to the EDFA case” and not at exactly the same number of round trips? Does this simply mean that the FOPA-PSA constellations were only measured for selected (not all) round trips, so that the closest among the measured round-trips was chosen for the third column?
 - e. Looking at Fig. 4(b), the reach for the PSA case at optimum power is 49 round-trips, while the reach for the PIA case at optimum power is 9 round-trips, yielding reach improvement of $49/9 = 5.44$. What procedure was used to arrive at the reach improvement number equal to 5.6?
 - f. What is the residual dispersion of the SMF span after the pre- and post-compensation? Is it 100% compensated? It seems that the amounts of total dispersion compensation in the PSA and PIA cases are the same, only pre-/post-compensation ratios are different.
 - g. How much delay between the signal and idler remains to be compensated by the VDL? How large is VDL loss?
 - h. What is the bandwidth of the OTF-30M-12S2 optical filter used at the receiver?
 - i. The “Methods” section mentions that the loop experiment makes injection-locking stable. What about the PZT controller? Does it have to

re-adjust its phase after every circulation or it operates on a much slower scale?

- j. What exactly does “implementation penalty” mean on line 361 of the “Methods” section? This is some DSP jargon that most readers are unfamiliar with.

9. Minor English / typo corrections:

- a. Line 96 near the top of page 2: should “single-span OPCs” be replaced by “single OPCs”?
- b. Line 163 on page 2: replace “constituted of” by “consisted of”.
- c. Replace 3 instances of “pump recovery-stage” on page 2 by either “pump recovery stage” or “pump-recovery stage”. Same applies to Fig. 2. Same applies to “power tuning-stage” and “transmission-stage”.
- d. Line 200 on page 3: replace “corresponds” by “correspond”.
- e. Line 211 on page 3: replace “lunch” by “launch”.
- f. Replace “EDFA- and FOPA-PIA-amplification” and “PSA-amplification” in the last paragraph of “Constellation diagrams” section on page 3 by “EDFA- and FOPA-PIA-based amplification” and “PSA-based amplification”, respectively. Same applies to Fig. 2 caption.
- g. Similarly, replace “EDFA-case”, “FOPA-PIA-case” and “PSA-case” in “Bit error rate measurements” section on page 3 and in Fig. 2 caption by “EDFA case”, “FOPA-PIA case” and “PSA case”, respectively.
- h. Replace “ps/(nm km)” by “ps/nm” on lines 322–325 of “Methods” section (4 instances).
- i. Add comma between “separated” and “and” on line 334 of “Methods” section.

Reviewer #2 (Remarks to the Author):

Review of manuscript "Long-haul optical transmission links using low-noise phase-sensitive amplifiers," by Samuel L.I. Olsson et al.

The manuscript presents an impressive transmission experiment showing the potential of a scheme based on a non-degenerate PSA to increase the transmission reach in a long-haul transmission system. Both noise and nonlinearity mitigation has been demonstrated. The manuscript is convincing, and I suggest publication in Nature Communications. I have a few comments that I would like the authors address before publication.

1. It is claimed that the technique is WDM compatible. WDM would require that a single pump is used for multiple channels (is it so?), and that the polarization of all the channels is aligned. Can the authors comment on the tolerance to polarization mode dispersion in the transmission fiber? Would this scheme lock the signal polarization and mitigate for the PMD as well?
2. The recovery of the signal polarization is a tricky part. Did the author consider using some of the polarization independent scheme that have been proposed in the past? Including polarization diversity schemes?
3. How this scheme compares with the scheme of ref. [27]? Is its performance expected to be higher, or different?
4. At the end of the link only the signal is detected, and the idler is dumped. Could in principle be possible to use the information contained in the idler to further reduce the noise? Or the gain (because of the strong correlation) would be insignificant?

Reviewer #3 (Remarks to the Author):

Phase sensitive amplifiers represent a promising technology for improving the transmission capacity in future optical fibre communication systems by enabling low noise signal transmission. Even though the technological challenges for their commercial deployment are enormous, some of them have been already highlighted in this paper, they are still an attractive research topic with many potentials for application in quantum communications, high speed digital transmission and other areas.

This paper comes from a research group which has pioneered in the PSA field. Over the last years they have presented very sophisticated PSA subsystems and they have demonstrated world record results. Having implemented experimentally a non-degenerate PSA link with the lowest noise figure performance, published in nature photonics [5], they now demonstrate its application on long haul transmission.

I would like to see this paper published, as I believe it is an important contribution to the field. On the other hand, I have some major concerns that I would like the authors to address in a convincing way.

The paper presents the results of a long-haul transmission experiment for a single channel QPKS signal in low noise PSA assisted transmission links and performs comparison with EDFA and FOPA-PIA assisted links. The core of this work, which involves the EDFA-PSA comparison, has been

already presented at the post-deadline sessions of ECOC in 2014 [9]. Although the results and conclusions of [9] do not seem to have been published in any other peer-reviewed journal, the large time lag between the first announcement and now may count against the impact of this publication. In addition, I would have expected this submission to include multi-channel transmission results, as well as, the use of more advanced signal formats. Despite the transparency in the PSA operation, it would be extremely interesting to quantify the transmission reach improvement that can be achieved in those cases when compared to EDFA/FOPA-PIA links.

The authors claim a 5.6 times reach improvement when using PSAs in the link instead of EDFAs. I am a deeply concerned about how fair this comparison is. The PSA link requires compensation of chromatic dispersion before each amplification stage. On the other hand, inline compensation of chromatic dispersion can be avoided in EDFA links with the use of electronic DSP equalization, thus enabling much longer transmission distances. This technology exists, and it is available in commercial systems, why ignoring it?

A couple of minor points.

In the introductory section, 2nd column, 2nd paragraph (lines 52-69), the referred publications correspond to experiments where the PSA subsystem acts as regenerator without achieving low-noise amplification. Please make this clarification.

In [9], there was a 3.3 times reach improvement of the PSA assisted transmission compared to the use of the EDFAs. In this submission, the number has been increased to 5.6, but only with a slight improvement in the PSA transmission reach. Furthermore, it seems that the maximum reach of the EDFA system has been decreed by almost 30%. Please clarify what exactly has happened in your experimental setup and why you have followed the specific design option.

We would first of all like to thank the reviewers for providing constructive comments on the manuscript. Our detailed response to the comments is presented in this letter. In addition to the changes that are made based on the reviewers' comments we have also made a few minor changes, mainly corrections of typos. All changes are indicated in the highlighted version of the manuscript.

Reviewer 1

The authors report a very thorough and systematic study of performance of an ultra-long-haul 10 Gbaud QPSK link employing phase-sensitive amplifiers (PSAs) in comparison to the links based on conventional phase-insensitive fiber-optic parametric amplifiers (FOPA-PIAs) or EDFAs with nearly quantum-limited noise figures. They demonstrate a very impressive result: the PSA-enabled link is capable of 5.6 times greater reach than the reach of the EDFA- or FOPA-PIA-based link. Even more remarkably, they show that the PSA-based link can tolerate a total accumulated nonlinear phase shift of 6.17 radians, incurred by each of signal and idler beams, which is 6-8 times greater than the nonlinear tolerance of the conventional PIA-based links.

It was previously predicted [e.g., *Opt. Express* **13**, 7563 (2005)] that the two-mode PSAs can improve the long-haul transmission links by combination of the reduction in the amplified spontaneous emission (ASE) noise and mitigation of nonlinear impairments. The authors present the first experiment that proves this prediction by showing that the net reach increase is noticeably greater than 4 times (the number expected from just the 6-dB ASE noise suppression in a linear propagation regime), i.e., both ASE reduction and nonlinear mitigation are fully realized here. These impressive results can influence the optical communication community to accept the PSAs as an attractive alternative to conventional amplifiers.

The key enablers of these original and impressive demonstrated results are the PSAs - a novel class of amplifiers rapidly gaining interest in the optical communication field. The authors reported preliminary observations of PSA- and PIA-based QPSK long-haul links at a conference [9], and this manuscript presents a much refined, very thorough, and definitive version of the PSA/PIA comparison, which sets a very high bar for proper "apple-to-apple" experimental comparisons between the links employing different amplifier types. These comparisons are extremely challenging, because the two-mode PSA is not just a simple drop-in replacement for an EDFA, especially in a recirculating loop, and I expect that the authors outstanding work will serve as a "gold standard" for how such comparative experiments should be carried out in the future.

The authors' conclusions are very convincing and supported by large amounts of well presented and clearly-explained data.

I strongly recommend this manuscript to be accepted for publication in *Nature Communications*. To improve the clarity of the presentation of this very sophisticated experiment, I advise the authors to consider the following suggestions / comments:

1. The authors should be careful in using statements about PSAs capable of adding 6 dB less noise than PIAs (3 occurrences: abstract, 1st paragraph of the introduction, last paragraph of "Basic principles") without qualifying / clarifying them. For the same signal power gain as a PIA, PSA indeed produces 6 dB less ASE power. This 6-dB advantage, however, is not "real," i.e., it does not result in 6-dB improvement in the signal-to-noise ratio (SNR) from PIA to PSA for the same amplifier input power. In the case of a signal/idler degenerate (one-mode) PSA, this is because the ASE after the PSA no longer has completely random phase (its phase is aligned with that of the amplified signal, i.e., the PSA quadrature noise is only 3-dB lower than that of the PIA), which leads to detected SNR improvement over PIA to be only 3 dB. In the case of a two-mode PSA, the ASE of the signal still has random phase, but the effective input signal power into the amplifier is doubled compared to the PIA, because of the presence of data-carrying idler beam that is, for the same total input power as the PIA the two-mode PSA yields only 3-dB better SNR. Hence, in the two-mode PSA the second 3 dB of the 6-dB noise reduction is an artefact of propagating twice as much total power in the transmission link.

Thus, I believe the reader is misled when the manuscript mentions 6-dB noise reduction without adding that it requires propagating twice as much total power in the transmission fiber. While PSA experts can understand what the authors mean, the general Nature Communications audience will be very confused by 6-dB noise reduction (as opposed to 3-dB reduction well-known from quantum mechanics [4]) without a proper mentioning of the corresponding power increase. Here are the suggested modifications for the three occurrences of “6 dB less noise”:

Response: We thank the reviewer for the in-depth discussion on how to discuss the noise figure difference between PIAs and PSAs and agree that some readers can be confused by the current statement that PSAs add 6 dB less noise compared to PIAs.

a. Abstract: a good way to explain what is responsible for the 6-dB less noise is by modifying the third sentence from the end of the abstract into “The phase-sensitively amplified link transmits two data-carrying waves, thus occupying twice the bandwidth **and propagating twice the total power** compared to the phase-insensitively amplified link.” (boldface is the added modification)

Response: The manuscript has been revised according to the reviewer’s suggestion.

b. In the second sentence of the introduction, “6 dB less noise” is very ambiguous, because neither “noise” nor “gain” is defined here. As discussed above, the 6-dB figure artefact comes from omitting the proper definition of either gain (not accounting for increased input power) or noise (not considering that all amplified noise power is concentrated in one quadrature). Citing Ref. [4] in this context is also misleading, because this reference never discussed more than 3 dB of PSA advantage over PIA. Thus, to avoid any confusion and controversy, the authors should rewrite the second sentence of the introduction in terms of the PSA noise figure, on which everyone agrees that it is lower than the ideal PIA noise figure by 3 dB: “Phase-sensitive amplifiers (PSAs) can provide low-noise amplification, **because at high gains their noise figure is 3 dB lower and compared to than that of even ideal phase-insensitive amplifiers (PIAs) operating at the same gain, PSAs add 6 dB less noise to the signal** [4], [5].”

Response: The manuscript has been revised according to the reviewer’s suggestion with some minor additions providing further clarification.

c. In the last paragraph of “Basic principles,” after the sentence ending with “...PSA will add 6 dB less noise than a PIA.” an explanation needs to be given, along the lines of “**First 3 dB of this improvement comes from the phase-sensitive nature of the gain, which releases the PSA from being constrained by the 3-dB quantum limit on PIA noise figure [4], at the expense of sacrificing half of the available bandwidth for propagating the idler [*]. This noise-figure improvement has been characterized in detail in [**,***]. The second 3 dB of the improvement comes from the fact that the data in the two-mode-PSA amplified link is carried by two beams (signal and idler) of equal powers, which makes the effective total data-carrying power in the PSA link twice that of the PIA link [*].**” Here Ref. [*] is Opt. Express **13**, 7563 (2005), in which the two-mode PSA and copier-PSA scheme were first proposed and explained, and [**,***] are Opt. Express **18**, 14820 (2010) and Opt. Lett. **36**, 722 (2011).

Response: The suggested references have been added and the suggested text has been added with minor modifications.

I would like to stress that the need to clarify the 6-dB improvement does not take anything away from the breakthrough experimental results demonstrated in the paper. Such clarification, in my opinion, is needed to give these results the physical interpretation they deserve, without contradicting quantum mechanics. From a practical perspective, the reader also needs to be made aware that there is “no free lunch”: two-mode PSA requires more bandwidth and more propagating power to achieve 6-dB better noise performance than PIA.

2. I suggest using singular word “link” in the title of the paper.

Response: The suggested change to the title has been made.

3. The authors use the term “span-wise nonlinearity mitigation” without explaining what it means. I think something like “**nonlinearity mitigation after each span**” would make it more clear.

Response: The manuscript has been revised according to the reviewer’s suggestion.

4. I would also caution authors against making explicit suggestions (in the abstract and conclusions) of potential use of the PSAs for entangled photon amplification. While PSAs are important for generation of squeezed and continuous-variable entangled states and could find some other potential applications in quantum information sciences, such as that in Ref. [39], they have been proven useless for quantum communications and detrimental to entangled states [“Gaussian-only regenerative stations cannot act as quantum repeaters,” R. Namiki, O. Gittsovich, S. Guha, and N. Ltkenhaus, Phys. Rev. A **90**, 062316 (2014)]. I suggest limiting the authors’ statements to a more general form such as “might also find applications in the field of quantum information science, where generation and processing of quantum states are of interest.”

Response: The text has been changed in both the suggested places to reflect the comment.

5. In addition to some of the references above, the manuscript could also benefit from a couple of more references, such as:

a. Citing [Science **348**, 1445 (2015)] in the context of “Cancellation of nonlinear distortion by digital signal processing” in the last paragraph of page 1;

Response: The suggested citation has been added.

b. Mentioning in the last sentence of the second-to-last paragraph of page 1 that the multi-channel compatible two-mode PSAs can be potentially combined with multi-channel amplitude regenerators [Nature Comm. **8**, 884 (2017)] for multi-channel regeneration of advanced modulation formats.

Response: The citation along with a new sentence has been added at the end of the paragraph.

6. 4th sentence of the “Discussion” section stating that “The performance in linear regime is not expected to depend on neither symbol rate or modulation format” should be rephrased as “The **copier-PSA** performance in linear regime is not expected to depend on neither symbol rate or modulation format”. Otherwise, “performance” implies BER, which of course will depend on both symbol rate and modulation format, and keeping same BER would require higher SNRs for higher symbol rates and more advanced formats.

Response: The manuscript has been revised according to the reviewer’s suggestion.

7. A couple of suggestions for the Figures:

a. “Colored arrows indicating waves” in Fig. 2 appear too small to be recognized as arrows (can be easily confused with circles denoting WDM couplers and input and received power points) perhaps they can be enlarged.

Response: The size of the arrows in Fig. 2 has been increased.

b. EDFA and PIA data in Fig. 4(a) would benefit from employing different symbols (e.g., filled vs empty circles) to show that there are actually two nearly overlapping data sets for each launch power.

Response: The figure has been revised to make a distinction between the EDFA and PIA case.

8. There are a number of small details about the experiment that need to be added or clarified, such as:

a. What is PRBS15? Is it 2^{15} , $2^{15} - 1$ or something else?

Response: It is a PRBS pattern of length $2^{15} - 1$ and the text has been changed to make this clear.

b. What does it mean “The received power was measured at point P_{rec} ” on line 182 of page 3? Does it mean that the signal going into the receiver EDFA (green box in Fig. 2) was attenuated by amount equivalent to the loss of “Polarization tuning and pump recovery-stage” unit, but this attenuator is not shown in Fig. 2?

Response: There was no attenuator before the receiver EDFA and the loss of the last span was therefore slightly lower (~ 2 dB) than the loss of the other spans. For the EDFA case and the FOPA-PIA case this resulted in slightly better performance compared to what would have been achieved in a link in which all spans had the same loss. For the PSA case the performance was still worse than it would have been if the last amplifier in the link was a PSA. Note however that the impact of having slightly lower loss in the last span is negligible after many circulations. Three sentences have been added in the “Methods” section to clarify this for the reader.

c. The important fact that in the PSA-based link the signal power launched into the first loop round-trip was 6 dB higher than that going into the second round-trip is not mentioned until the Methods section at the end of the article. I think this information should be provided much earlier - either in “Experimental set-up” or in “Constellation diagrams” section. Otherwise, this lack of information leaves the reader puzzled throughout most of the manuscript about whether the signal power in the SMF in the first loop is 6-dB lower than in the subsequent loops or not.

Response: We agree that this needs to be made clear earlier in the manuscript. A sentence has been added to the last paragraph of “Experimental set-up” to clarify this.

d. Why does the third column of Fig. 3 show “the constellations for the FOPA-PSA case at the closest available number of round trips to the EDFA case” and not at exactly the same number of round trips? Does this simply mean that the FOPA-PSA constellations were only measured for selected (not all) round trips, so that the closest among the measured round-trips was chosen for the third column?

Response: That is correct, for the PSA case we did not measure every round trip. To make the measurement and data acquisition process faster we used the FastFrame(TM) Segmented Memory feature of the receiver oscilloscope together with a trigger signal that we could set to trigger the scope on every round trip, every other round trip, or every third round trip. For the PSA case, where we were interested in capturing many round trips, we could not trigger on every round trip due to memory limitations in the receiver scope and instead we had to trigger on and capture every third round trip, i.e., round trip 1, 4, 7, 10, 13, ..., 49.

e. Looking at Fig. 4(b), the reach for the PSA case at optimum power is 49 round-trips, while the reach for the PIA case at optimum power is 9 round-trips, yielding reach improvement of $49/9 = 5.44$. What procedure was used to arrive at the reach improvement number equal to 5.6?

Response: When calculating the reach improvement factor we interpolated between the measurement points to obtain the reach corresponding to a BER of exactly 10^{-3} . For the EDFA case the optimum launch power was 4 dBm and the number of round trips for a BER of 10^{-3} was 8.73. For the PSA case the optimum launch power was 6 dBm and the number of round trips for a BER of 10^{-3} was 48.81. The reach improvement was calculated as $48.81/8.73 = 5.59 \approx 5.6$.

f. What is the residual dispersion of the SMF span after the pre- and post-compensation? Is it 100% compensated? It seems that the amounts of total dispersion compensation in the PSA and PIA cases are the same, only pre-/ post-compensation ratios are different.

Response: The amount of per span residual dispersion was experimentally optimized for longest reach in a nonlinear transmission regime for the PSA case and was less than 35 ps/nm. This amount of residual dispersion had a negligible impact on the performance in linear transmission regime both for the PSA and PIA cases as well as for the PIA cases in nonlinear transmission regime due to the low symbol rate and few round trips. The total amount of dispersion compensation was the same in the PSA and PIA cases. Two

sentences has been added to the “Methods” section discussing the amount of residual dispersion.

g. How much delay between the signal and idler remains to be compensated by the VDL? How large is VDL loss?

Response: The total delay between the signal and idler was approximately 10 ns. Most of this delay was compensated for by a patch cord and the remaining delay was compensated using the VDL. The insertion loss of the VDL was 1 dB. The sentence about the VDL in the “Methods” section has been changed to include the insertion loss.

h. What is the bandwidth of the OTF-30M-12S2 optical filter used at the receiver?

Response: The 3 dB bandwidth of the optical filter is 0.9 nm. The sentence in the “Methods” section describing the optical filter has been changed to include the filter bandwidth.

i. The “Methods” section mentions that the loop experiment makes injection-locking stable. What about the PZT controller? Does it have to re-adjust its phase after every circulation or it operates on a much slower scale?

Response: The PZT-based PLL was tracking changes in the relative phase between the signal, the idler, and the pump *within* the transmission loop. Changes in the relative phase can be introduced by thermal drift and acoustic noise in the sections where the waves are not propagating in the same fiber, i.e., in the power tuning stage and in the polarization tuning and pump recovery stage and are typically on a time-scale of up to a couple of hundred Hertz. Since the PLL is only tracking changes within the loop it does not have to re-adjust its phase after each circulation but will instead continuously adjust its phase over time to accommodate the changes in the relative phase between the waves.

j. What exactly does “implementation penalty” mean on line 361 of the “Methods” section? This is some DSP jargon that most readers are unfamiliar with.

Response: We agree that some readers might find this expression unfamiliar. The sentence has been changed to make it more understandable for someone not familiar with the term.

9. Minor English / typo corrections:

a. Line 96 near the top of page 2: should “single-span OPCs” be replaced by “single OPCs”?

b. Line 163 on page 2: replace “constituted of” by “consisted of”.

c. Replace 3 instances of “pump recovery-stage” on page 2 by either “pump recovery stage” or “pump-recovery stage”. Same applies to Fig. 2. Same applies to “power tuning-stage” and “transmission-stage”.

d. Line 200 on page 3: replace “corresponds” by “correspond”.

e. Line 211 on page 3: replace “lunch” by “launch”.

f. Replace “EDFA- and FOPA-PIA-amplification” and “PSA-amplification” in the last paragraph of “Constellation diagrams” section on page 3 by “EDFA- and FOPA-PIA-based amplification” and “PSA-based amplification”, respectively. Same applies to Fig. 2 caption.

g. Similarly, replace “EDFA-case”, “FOPA-PIA-case” and “PSA-case” in “Bit error rate measurements” section on page 3 and in Fig. 2 caption by “EDFA case”, “FOPA-PIA case” and “PSA case”, respectively.

h. Replace “ps/(nm km)” by “ps/nm” on lines 322-325 of “Methods” section (4 instances).

i. Add comma between “separated” and “and” on line 334 of “Methods” section.

Response: We thank the reviewer for the attentive reading of the manuscript. All of the above listed typos have been corrected.

Reviewer 2

The manuscript presents an impressive transmission experiment showing the potential of a scheme based on a non-degenerate PSA to increase the transmission reach in a long-haul transmission system. Both noise and nonlinearity mitigation has been demonstrated. The manuscript is convincing, and I suggest publication in Nature Communications. I have a few comments that I would like the authors address before publication.

1. It is claimed that the technique is WDM compatible. WDM would require that a single pump is used for multiple channels (is it so?), and that the polarization of all the channels is aligned. Can the authors comment on the tolerance to polarization mode dispersion in the transmission fiber? Would this scheme lock the signal polarization and mitigate for the PMD as well?

Response: That is correct, a single pump would be used for multiple channels and the polarization of all channels entering the PSA should be aligned for optimum performance. For details on polarization alignment and tracking, see [A. Lorences-Riesgo et al. Journal of Lightwave Technology vol. 34, no. 13, pp. 3171-3180, (2016)]. A sentence has been added to the third paragraph in the introduction referencing that paper and mentioning the alignment of polarization.

Through its polarization dependent gain the PSA will realign the polarization of the signals and to some extent mitigate the effect of PMD. However, there will be a penalty associated with the PMD mitigation that grows larger with the amount of PMD present. We have not investigated this effect and are thus unable to provide any quantitative estimate on the PMD tolerance. The question is very interesting and it is a good topic for future studies.

2. The recovery of the signal polarization is a tricky part. Did the author consider using some of the polarization independent scheme that have been proposed in the past? Including polarization diversity schemes?

Response: Due to the complexity increase associated with using a polarization independent scheme we decided not to aim for polarization independent operation in this particular experiment but instead focus on the intrinsic low-noise amplification and nonlinearity mitigation properties of the PSA.

3. How this scheme compares with the scheme of ref. [27]? Is its performance expected to be higher, or different?

Response: Our scheme is different from the scheme in [27] in several important ways. First of all, with in-line PSAs, we will not only benefit from nonlinearity mitigation but also from low-noise amplification with a theoretical quantum limited noise figure of 0 dB at high gains. In [27] the transmission link is amplified using in-line EDFAs for which the quantum limited noise figure is 3 dB. With in-line PSAs the nonlinearity mitigation is performed all-optically after each transmission span and we believe that there can be benefits to performing the nonlinearity mitigation several times along the transmission link compared to only once in the receiver DSP as is done in [27]. Moreover, using in-line PSAs we are not only able to mitigate deterministic nonlinear distortions originating from signal-signal interaction but also the non-deterministic distortions originating from signal-noise interaction. Mitigation of non-deterministic distortions is not possible using the scheme in [27]. An advantage of the scheme in [27] is that in-line dispersion compensation is not required which can be beneficial in a nonlinear transmission regime. We are not aware of any exhaustive direct comparison between the scheme in [27] and our scheme but based on the differences listed above we expect the performance to differ.

4. At the end of the link only the signal is detected, and the idler is dumped. Could in principle be possible to use the information contained in the idler to further reduce the noise? Or the gain (because of the strong correlation) would be insignificant?

Response: Due to the strong correlation between the two waves the gain from detecting both signal and idler after a PSA are in general insignificant and originates only from reduction in

noise added by the receiver in the electrical domain with two parallel receivers.

In our experiment, where the last amplifier in the PSA case was a PIA, we would however expect a performance improvement by detecting both signal and idler followed by coherent addition in DSP. This performance improvement would originate from improved noise performance and from mitigation of deterministic nonlinear distortions originating from signal-signal interaction in the last transmission span.

Reviewer 3

Phase sensitive amplifiers represent a promising technology for improving the transmission capacity in future optical fibre communication systems by enabling low noise signal transmission. Even though the technological challenges for their commercial deployment are enormous, some of them have been already highlighted in this paper, they are still an attractive research topic with many potentials for application in quantum communications, high speed digital transmission and other areas.

This paper comes from a research group which has pioneered in the PSA field. Over the last years they have presented very sophisticated PSA subsystems and they have demonstrated world record results. Having implemented experimentally a non-degenerate PSA link with the lowest noise figure performance, published in nature photonics [5], they now demonstrate its application on long haul transmission.

I would like to see this paper published, as I believe it is an important contribution to the field. On the other hand, I have some major concerns that I would like the authors to address in a convincing way.

The paper presents the results of a long-haul transmission experiment for a single channel QPKS signal in low noise PSA assisted transmission links and performs comparison with EDFA and FOPA-PIA assisted links. The core of this work, which involves the EDFA-PSA comparison, has been already presented at the post-deadline sessions of ECOC in 2014 [9]. Although the results and conclusions of [9] do not seem to have been published in any other peer-reviewed journal, the large time lag between the first announcement and now may count against the impact of this publication. In addition, I would have expected this submission to include multi-channel transmission results, as well as, the use of more advanced signal formats. Despite the transparency in the PSA operation, it would be extremely interesting to quantify the transmission reach improvement that can be achieved in those cases when compared to EDFA/FOPA-PIA links.

Response: We agree with the reviewer that multi-channel transmission as well as the use of more advanced modulation formats are very interesting areas to explore. However, due to the complexity and challenges associated with performing long-haul transmission experiments using two-mode PSAs we believe that the results presented in this manuscript already presents a significant leap forward compared to prior journal publications and thus justify publication in Nature Communications. Our manuscript also presents a substantial extension of the work that was presented at the post-deadline session of ECOC 2014 and important improvements have been made to the experimental setup which allows us to demonstrate a reach improvement factor of 5.6 which should be compared to our previously demonstrated reach improvement factor of three.

The authors claim a 5.6 times reach improvement when using PSAs in the link instead of EDFAs. I am a deeply concerned about how fair this comparison is. The PSA link requires compensation of chromatic dispersion before each amplification stage. On the other hand, inline compensation of chromatic dispersion can be avoided in EDFA links with the use of electronic DSP equalization, thus enabling much longer transmission distances. This technology exists, and it is available in commercial systems, why ignoring it?

Response: It is correct that PSA-amplified transmission links require in-line dispersion compensation while this is not required for EDFA-amplified transmission links. The reason why we decided to compare the PSA-amplified transmission link to an EDFA-amplified link with

in-line dispersion compensation was that we wanted an as straight forward and easy comparison as possible between the two amplification schemes. By keeping the dispersion compensation as well as, e.g., the span loss, the same for all schemes we are able to clearly attribute the reach improvement obtained with PSA-based amplification to the low-noise amplification and nonlinearity mitigation capabilities of the PSA. Performance improvements can likely be obtained for all considered amplification schemes by reducing span loss and further optimizing the experimental setup. An additional argument in favor of comparing dispersion compensated links is that many existing legacy system are optically compensated, and there is an industrial interest in upgrading such systems without replacing the optical infrastructure. Thus it is relevant to compare with EDFA-only and maintained dispersion map.

We also want to comment on the possible performance benefit of using electronic dispersion compensation that the reviewer is referring to. Assuming that the span loss is unchanged we do not expect any significant performance improvement in our particular transmission system for the EDFA case by using electronic dispersion compensation instead of in-line dispersion compensation. This may be unexpected but it is important to note that in our system:

1. We optimize the span dispersion map for the in-line compensated EDFA link.
2. We are transmitting a single data channel as opposed to multiple channels.
3. We are operating at 10 GBd where the dispersive length \gg span length.

The GN model [Poggiolini, P. “The GN model of non-linear propagation in uncompensated coherent optical systems.” *Journal of Lightwave Technology* 30.24 (2012) pp. 3857-3879] predicts better performance in uncompensated links compared to in-line compensated links but that is for the case with many WDM channels, high symbol rates, and long transmission distances.

See also [Mohammad S. Alfiad et al. “A comparison of electrical and optical dispersion compensation for 111-Gb/s POLMUX-RZ-DQPSK,” *Journal of Lightwave Technology*, Vol. 27, No. 16, pp. 3590-3598 (2009)] for a comparison between in-line dispersion compensation and electronic dispersion compensation in DSP. In that paper they show in experiments that when assuming an optimized dispersion map and some small amount of residual dispersion, the in-line dispersion compensated and the electronically dispersion compensated link perform similarly.

A couple of minor points.

In the introductory section, 2nd column, 2nd paragraph (lines 52-69), the referred publications correspond to experiments where the PSA subsystem acts as regenerator without achieving low-noise amplification. Please make this clarification.

Response: The manuscript has been revised to reflect the comment.

In [9], there was a 3.3 times reach improvement of the PSA assisted transmission compared to the use of the EDFAs. In this submission, the number has been increased to 5.6, but only with a slight improvement in the PSA transmission reach. Furthermore, it seems that the maximum reach of the EDFA system has been decreed by almost 30%. Please clarify what exactly has happened in your experimental setup and why you have followed the specific design option.

Response: When comparing the transmission reaches presented in this manuscript to the results presented in [9] it is important to note that the span loss in [9] was 38 dB and that the span loss for this work was 39 dB (specified in the last paragraph of the experimental setup section). This explains why the number of round trips is reduced for the EDFA case from 11 in [9] to 9 in this work. This is a reduction of about 20% which is close to the approximate expected reduction of about 25% (from $1 - 10^{(39-38)/10}$). Despite the fact that the span loss is increased by 1 dB the number of round trips in the PSA case is increased from 33 in [9] to 49 in this work, an increase of almost 50%.

Several changes were made to the experimental setup. The most important change was that we used the same laser to both generate and injection-lock the pump, as opposed to generating the pump with one laser and injection-lock it with another. This drastically improved the stability and performance for the PSA-amplified link by reducing pump degradation due to frequency drift between the master and slave lasers. We used tunable FBG DCMs for dispersion compensation instead of a combination of DCF and fixed FBG DCMs which was used in [9]. By using tunable FBG DCMs we were able to experimentally optimize the dispersion map for each amplifier configuration. This was not possible in [9]. We also added a polarization tuning stage that allowed for individual polarization alignment of the signal and idler. In [9], individual tuning of signal and idler polarization before the PSA was not possible. These changes were made to provide better tuning capabilities which in turn provided better performance.

Best regards,
Samuel Olsson with co-authors

REVIEWERS' COMMENTS:

Reviewer #1 (Remarks to the Author):

I am satisfied with the revisions performed by the authors and recommend this manuscript for publication in Nature Communications.

I only have one tiny suggestion the authors may want to consider: in Fig. 4b the traces of EDFA and PIA completely overlap, and it is not obvious from the figure that one set of PIA empty-circle symbols is buried under another set of EDFA filled-circle symbols. By using slightly larger size of empty circles compared to the size of filled circles, the authors could make both sets of symbols visible in the Figure, eliminating any readers' doubts.

Reviewer #2 (Remarks to the Author):

The authors satisfactorily addressed my comments in the original version of the manuscript. The paper can be accepted for publication.

In this letter we present our response to the final comment from Reviewer #1.

Reviewer 1

I am satisfied with the revisions performed by the authors and recommend this manuscript for publication in Nature Communications.

1. I only have one tiny suggestion the authors may want to consider: in Fig. 4b the traces of EDFA and PIA completely overlap, and it is not obvious from the figure that one set of PIA empty-circle symbols is buried under another set of EDFA filled-circle symbols. By using slightly larger size of empty circles compared to the size of filled circles, the authors could make both sets of symbols visible in the Figure, eliminating any readers' doubts.

Response: The empty-circle symbols both in Fig. 4(a) and in Fig. 4(b) have been enlarged and the EDFA case and the PIA case can now easily be distinguished.

Best regards,
Samuel Olsson with co-authors